# Kaposi Sarcoma in a Child after Fanconi Anemia-Induced Haploidentical Hematopoietic Stem Cell Transplant: A Case Report

**DOI:** 10.3390/children10020188

**Published:** 2023-01-19

**Authors:** Mohammed Saud Alsaidan, Ohoud Zaid Aljarbou, Waleed Alajroush

**Affiliations:** 1Internal Medicine Department, College of Medicine, Prince Sattam Bin Abdulaziz University, Al Kharj 11942, Saudi Arabia; 2Consultant Pathologist, Pathology and Laboratory Medicine, King Abdulaziz Medical City, Ministry of National Guard-Health Affairs, College of Medicine, King Saud Bin Abdulaziz University for Health Sciences, Riyadh 14611, Saudi Arabia; 3Pediatric Dermatology, King Abdullah Specialized Children’s Hospital, Dermatology, King Abdulaziz Medical City, Ministry of National Guard-Health Affairs, Riyadh 14611, Saudi Arabia

**Keywords:** kaposi sarcoma, stem cell transplant, fanconi anemia, Saudi Arabia

## Abstract

Kaposi sarcoma is relatively common after solid organ transplantation, but very rare after hematopoietic stem cell transplant (HSCT). Here we are reporting a rare case of Kaposi sarcoma in a child after HSCT. An 11-year-old boy with Fanconi anemia was treated by haploidentical HSCT from his father. Three weeks after transplantation, the patient developed severe graft-versus-host disease (GVHD) which was treated by immunosuppressive therapy and extracorporeal photopheresis. Approximately 6.5 months after HSCT, the patient had asymptomatic nodular skin lesions over the scalp, chest, and face. Histopathological examination showed typical findings of Kaposi sarcoma. Later, additional lesions in the liver and oral cavity were confirmed. Liver biopsy was positive for HHV-8 antibodies. The patient was continued on Sirolimus which was already being used for the treatment of GVHD. Cutaneous lesions were also treated with topical timolol 0.5% ophthalmic solution. Within six months, cutaneous and mucous membrane lesions were completely resolved. Follow-up abdominal ultrasound and MRI showed the disappearance of the hepatic lesion.

## 1. Introduction

Kaposi sarcoma (KS) is an angio-proliferative tumor characterized by spindle-shaped vascular cells, typically initiated by an infection with human herpesvirus-8 (HHV8), immunosuppression, or immunodeficiency [1]. However, HHV-8 infection alone does not necessarily cause KS [1]. Other factors including genetic, immunological, and environmental factors contribute to the development of KS [2,3]. It is a multifocal disease affecting mainly the skin and to a lesser extent the oral cavity, gastrointestinal tract, lymph nodes, and lungs [1]. Epidemiologically, Kaposi sarcoma has four different types: sporadic (classic), endemic (African), epidemic (AIDS-related), and iatrogenic (transplant-related) [4]. The latter is relatively common after solid organ transplantation, especially the kidney, liver, and heart [5,6]. It affects up to 1.5% of solid organ recipients [5,6]. However, Kaposi sarcoma is very rare after hematopoietic stem cell transplant (HSCT) [7]. The risk of post-transplantation Kaposi sarcoma is probably caused by prolonged use of immunosuppressive therapy after transplantation, which may ultimately reactivate HHV8 infection [1,8]. Additionally, the majority of frequently used immunosuppressive therapies are pro-carcinogenic [9]. Here, we report a rare case of Kaposi sarcoma in a child after HSCT.

## 2. Case Report

An 11-year-old boy with Fanconi anemia (FANCA gene mutation) was treated by haploidentical HSCT from his father. The patient had no HLA match with any of his tested family members. The patient was seronegative for human immunodeficiency virus (HIV) and HHV8 tests. The father was not tested for HHV-8. The conditioning regimen was the following:Fludarabine 50 mg injection, vial (26 mg IV Piggyback over 1 h for 1 day) injected every 24 h on day (-7), (-6), (-5), (-4) and (-3)ATG (Genzyme) anti-human T-lymphocyte immune globulin ATG rabbit, 25mg inj (10 mg IV Syringe infused as protocol) with concentration of 0.33 mg/mL, injected on day (-5) and day (-4)

For graft-versus-host disease (GVHD) prophylaxis, the patient received cyclophosphamide, two doses of 500 mg intravenously, with 80 mL of dextrose 5% in water over one hour, on day + 3 and day + 4, while Cyclosporine was started on day + 5, then mycophenolate mofetil was also added but Cyclosporine was changed to Tacrolimus on day + 12 due to severe headache. Three weeks after transplantation, the patient developed a biopsy-proven GVHD that involved the skin (acute GVHD stage 1) and later the gut (acute GVHD stage 1). The overall clinical acute GVHD grade was grade two [10]. The patient was started on corticosteroids (Prednisolone 1 mg/kg/day and Budesonide) and was also started on antitumor kinase inhibitors (Sirolimus and Ruxolitinib) since studies showed the efficacy of lower doses of prednisone of 1 mg/kg/day or even 0.5 mg/kg/day [11]. As the GVHD did not improve with the above medications, the patient was admitted and additionally started on extracorporeal photopheresis (ECP). Post-transplant chimerism results are mentioned in Table 1.

During hospitalization and after two months from the HSCT, abdominal ultrasound showed small nodular lesions in the liver, which were not accessible for liver biopsy. The lesions were thought to be fungal and treated with multiple courses of antifungal medications (voriconazole) but without improvement. Magnetic resonance imaging (MRI) showed signal characteristics favoring hepatic regenerative/dysplastic nodules rather than a fungal infection. Additionally, the patient received multiple systemic medications to treat various conditions during hospitalization. These included phosphonomethanoic acid (foscarnet) and cytomegalovirus (CMV) immunoglobulins for PCR-confirmed CMV infection, metronidazole, and then vancomycin for PCR-confirmed clostridium difficile infection, cidofovir for hematuria with positive human polyomavirus 1 (BK virus), cefepime and amikacin for fever of unknown origin, insulin for steroid-induced hyperglycemia, and amlodipine for high blood pressure.

Approximately 7.5 months after HSCT, the patient was referred to dermatology during his admission. At that time, the patient had a one-month history of asymptomatic nodular lesions over the scalp, chest, and face (Figure 1). The lesions started first on the scalp, which was followed a few weeks later by similar lesions in the chest and face. On examination, the patient was conscious, well, and vitally stable. There were multiple violaceous firm purpuric nodules over the face, scalp, and chest. There were no lesions in mucous membranes nor genital areas. The differential diagnosis included leukemia cutis, pseudolymphoma, cutaneous fungal or bacterial infection, nodular vasculitis, and Kaposi sarcoma. Two skin biopsies were taken from the lesions; the first was taken from the chest nodule and the second was taken from the scalp nodule. For the first biopsy, histopathological examination showed typical findings of Kaposi sarcoma while the direct immunofluorescence (DIF) examination was negative for deposits of immunoglobulin G (IgG), IgA, IgM, and C3. For the second biopsy, bacterial, fungal, and mycobacterial cultures were performed with negative results.

Further investigations were then started to clarify the non-improved liver lesions. Positron emission tomography (PET) scan showed multiple hypermetabolic small hepatic foci in a background of heterogeneous liver uptake, which raised the suspicion of liver involvement with Kaposi sarcoma. A liver biopsy was then performed, and the liver tissue showed focal intersecting fascicles of uniform spindle cells and intervening blood-filled spaces between spindle cells. Additionally, the tissue was positive for HHV-8 antibodies (13B10 clone), which is consistent with Kaposi sarcoma. Furthermore, it was positive for CD31 antibodies (JC70A clone), CD34 antibodies (Qbend/10 clone), and smooth muscle actin (SMA, 1A4 clone) but negative for Epstein–Barr virus (RNA in situ hybridization). Two months later, the patient developed erythematous flat patches to the right of the base of the frenulum, the mid-soft palate, and behind the upper left third molar, suggestive of mucous membrane involvement with Kaposi sarcoma.

For the cutaneous lesions of Kaposi sarcoma, the patient was started on topical timolol 0.5% ophthalmic solution (with two drops applied three times daily over the skin nodules, not exceeding 10 drops total per application). The patient was continued on Sirolimus which was already being used for the treatment of GVHD. The patient was also on oral propranolol for treating high blood pressure which could helped in treating both cutaneous and hepatic Kaposi sarcoma. With topical treatments (as adjunctive treatments), the skin nodules started to flatten, and the mucous membrane lesions started to disappear. Within six months, the topical medications were discontinued with complete resolution of skin and mucous membrane lesions. Follow-up abdominal ultrasound and MRI showed the disappearance of the hepatic lesion. The patient had a good response to ECP and Sirolimus with the disappearance of gut GVHD and significant improvement of the skin GVHD. After that, ECP was discontinued and Sirolimus has been tapered off.

## 3. Discussion

We are reporting the case of a child with Kaposi sarcoma after haploidentical HSCT. Kaposi sarcoma is a rare complication of HSCT, and a recent report estimated its occurrence at a rate of 0.11% (one in 10,000 HSCT recipients) [7]. The total number of similar cases reported over the last three decades is approximately 32 cases [7]. Additionally, the current case report is considered an infrequent presentation of this rare complication. For example, this is probably the first case in a child with Fanconi anemia treated by HSCT. Children represent less than 30% of previously reported post-HSCT Kaposi sarcoma cases [7]. In contrast with the current case, previously reported post-HSCT Kaposi sarcoma in children had frequent lymph node involvement but no liver involvement [7]. Lastly, haploidentical HSCT from the father was previously reported in only two children [12,13]. Nevertheless, the current case report shared several characteristics that have been associated with post-HSCT Kaposi sarcoma: non-autologous HSCT, chronic GVHD, multifocal presentation, skin and oral cavity involvement, and short duration between HSCT and diagnosis of Kaposi sarcoma [7]. Skin lesions appeared in the current patient 6.5 months from HSCT compared to a median of 7 to 8 months in the Cesaro series and literature review [7].

The use of immunosuppressant drugs for the prevention of GVHD is known to be a predisposing factor for KS. A study reported that 91% of patients with Kaposi sarcoma in organ transplant recipients were seropositive for HHV-8 before transplantation [14]. Likewise, seronegative recipients can be infected with seropositive allograph; however, caution should be taken to detect seropositive by using different assays to reduce false negativity. Studies have also shown that KS didn’t develop in posttransplant seropositive patients until the immunosuppression was started, suggesting the reactivation of the virus [14,15]. Although testing is helpful for the prediction of KS in the recipient, it is also challenging as there is no validated serology kit that is commercially available [16]. Recent technologies, metagenomic next-generation sequencing (NGS) and whole-exome sequencing (WES) have been utilized for the early detection of HHV-8 and other infectious disease loads [17].

HHV-8 infection post-HSCT is common; however, Kaposi sarcoma with visceral involvement in the pediatric population is rare. In our case, the liver lesion of Kaposi sarcoma was initially misdiagnosed and treated as a fungal infection. The accurate diagnosis was confirmed only after the appearance of skin lesions suggestive of Kaposi sarcoma. Although this may reflect the unexpected clinical presentation, the findings underscore the importance of keeping Kaposi sarcoma in the differential diagnosis of post-HSCT complications. Further, KS was also reported to involve the lungs [12], tonsils, hard palate, paranasal sinuses [13], oral cavity, lymph node, and gut [3]. Another paper by Avivi et al., 2011, reported a case of KS in a young patient (46-years-old) post T-cell depleted allogeneic stem cell transplant for chronic myeloid leukemia [18]. The author reported that the patient did not receive any prophylaxis for GVHD, however, the patient developed grade II GVHD after four months, and therefore low dose prednisolone therapy was started. In our case, the patient developed GVHD after three weeks. The first sign of KS in the case report of Avivi et al. appeared after six months as nodular lesions on the patient’s trunk, legs, and oral cavity followed by respiratory nodules and liver failure with multiple hepatic nodules. Biopsies of lesions showed disseminated visceral KS. Interferon, low-dose doxorubicin, and broad-spectrum antibacterial and antifungal medication were given to patient, but despite all these therapies, the patient did not survive. In contrast, in our case, initially the patient developed a hepatic nodular lesion after two months, and later after six months, skin lesions appeared.

The management of KS includes treatment with chemotherapy according to disease severity and disseminated visceral involvement [3], reduction, change, or discontinuation of immunosuppressant agents which showed significant improvement in our case. However, the skin and mucous membranes lesions in the current patient also responded well to topical treatments in addition to Sirolimus. The current patient was not treated by the withdrawal of immunosuppressive therapy as the antitumor kinase inhibitors such as Sirolimus have anti-Kaposi sarcoma effects [19]. Furthermore, Sirolimus can be used in the treatment of cutaneous KS in post-solid organ transplant patients [3]. Consistent with the current positive outcome, children with post-HSCT Kaposi sarcoma have a good prognosis despite visceral involvement. For example, mortality was 11% in children and 48% in adults in the Cesaro series and literature review [7].

## 4. Conclusions

In conclusion, Kaposi sarcoma with visceral involvement is a rare secondary malignancy in the pediatric population after haploidentical HSCT as compared to solid organ transplant. The delay in diagnosis and delayed treatment can lead to increased mortality. Therefore, we present this case report to emphasize the importance of suspecting Kaposi sarcoma in patients post-HSCT. The unexpected involvement of the liver was initially misdiagnosed as a fungal infection but was suspected after the appearance of skin lesions. Despite visceral involvement, the patient responded well to topical treatment and systemic antitumor kinase inhibitors. The findings underscore the importance of keeping Kaposi sarcoma in the differential diagnosis of post-HSCT complications.

## Figures and Tables

**Figure 1 children-10-00188-f001:**
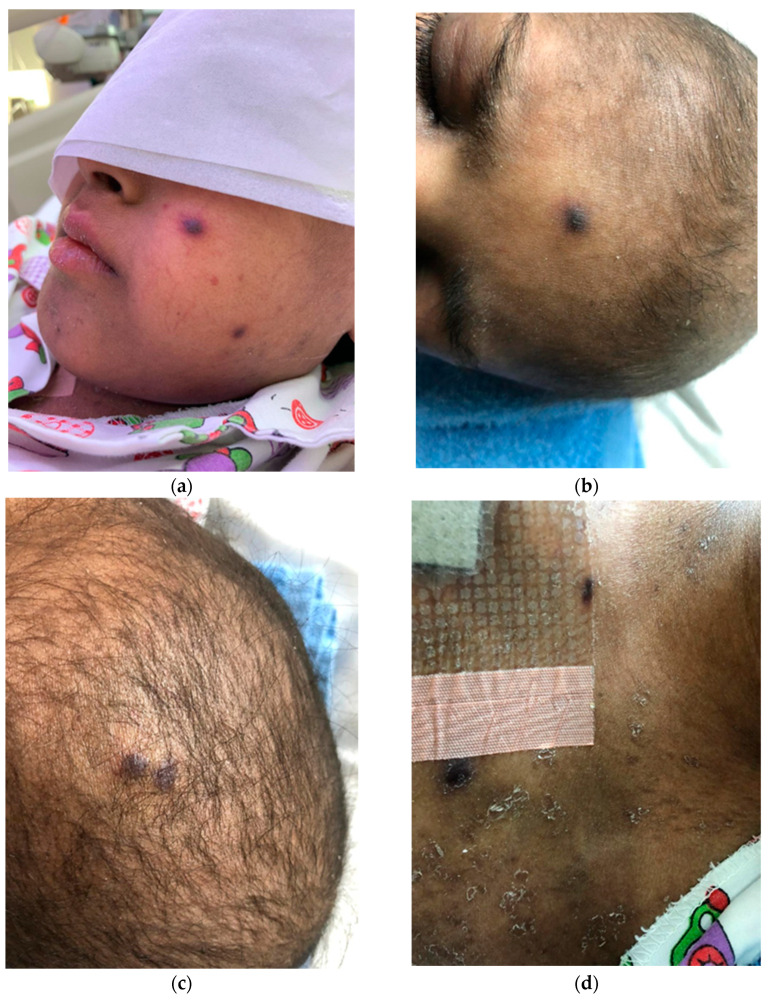
Distribution of skin lesions of Kaposi sarcoma in cheek (**a**), forehead (**b**), scalp (**c**), and chest (**d**).

**Table 1 children-10-00188-t001:** Post-transplant chimerism results engraftment.

	Day 39	Day 60	Day 172
Donor T-lymphocyte	100% (CV = 0%)	99.19% (CV = 1%)	98.07% (CV = 2%)
Donor Myeloid Cells	100% (CV = 0%)	98.78% (CV = 2%)	98.56% (CV = 2%)
Donor whole cells population	100% (CV = 0%)	98.83% (CV = 2%)	96.68% (CV = 6%)

A coefficient of variance (CV) ≤ 5% should be obtained with a minimum of three Short Tandem Repeats (STR) markers used in the overall % chimerism calculation.

## Data Availability

Not applicable.

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
