# Peer review of "Kaposi Sarcoma in a Child after Fanconi Anemia-Induced Haploidentical Hematopoietic Stem Cell Transplant: A Case Report"

_children, 2023, doi:10.3390/children10020188_

Round 1

Reviewer 1 Report

the paper by Alsaidan et al. is a case report of a case of Kaposi sarcoma involving the skin, liver, and oral cavity in a child with Fanconi anemia treated with haploidentical HSCT. With this paper, the authors underline the importance of keeping Kaposi sarcoma, even if rare, in the differential diagnosis of post-HSCT complications in the pediatric population. Moreover, even if other reports of Kaposi sarcoma after HSCT in children exist, this is the first one to describe liver involvement. The paper is generally well-written and the content is interesting, the case is well-reported, and similarities and differences from other cases of Kaposi sarcoma after HSCT present in literature are well illustrated.

However, hereunder are listed some minor remarks:

-       Line 32: I would suggest better detailing the expression “immune problem”

-       I would suggest adding the grade of aGvHD and discussing better why Prednisone was started at 1 mg/kg/day rather than 2 mg/kg/day which represents the gold standard. I would also recommend adding the response of aGvHD after extracorporeal photopheresis.

-       In the case description, I would suggest specifying which conditioning regimen was used and mentioning whether the patient’s father was seropositive or seronegative for HHV8. In the case of a seronegative father, I would suggest discussing the possible ways of infections and possible preventive measures.

-       I would suggest briefly discussing the possible options for the treatment of Kaposi sarcoma after HSCT and when chemotherapy should be used.

-       I would recommend mentioning what other organs have been described to be involved in Kaposi sarcoma in addition to those found in this patient. I would also suggest discussing the case by Avivi et al. (2021, Leuk Lymphoma) in which a liver involvement was described and underlying the differences and similarities with the reported case.

Author Response

Dear Reviewer, 

Thank you for your valuable feedback. We have done all the suggested changes in the manuscript highlighted in yellow. 

Reviewer 1

the paper by Alsaidan et al. is a case report of a case of Kaposi sarcoma involving the skin, liver, and oral cavity in a child with Fanconi anemia treated with haploidentical HSCT. With this paper, the authors underline the importance of keeping Kaposi sarcoma, even if rare, in the differential diagnosis of post-HSCT complications in the pediatric population. Moreover, even if other reports of Kaposi sarcoma after HSCT in children exist, this is the first one to describe liver involvement. The paper is generally well-written and the content is interesting, the case is well-reported, and similarities and differences from other cases of Kaposi sarcoma after HSCT present in literature are well illustrated.

However, hereunder are listed some minor remarks:

-       Line 32: I would suggest better detailing the expression “immune problem”

Changes have been made.

-       I would suggest adding the grade of aGvHD and discussing better why Prednisone was started at 1 mg/kg/day rather than 2 mg/kg/day which represents the gold standard. I would also recommend adding the response of aGvHD after extracorporeal photopheresis.

Thank you for the suggestion, these have been added in the case report.

-       In the case description, I would suggest specifying which conditioning regimen was used and mentioning whether the patient’s father was seropositive or seronegative for HHV8. In the case of a seronegative father, I would suggest discussing the possible ways of infections and possible preventive measures.

Thank you for the suggestion, these have been added in the case description.

-       I would suggest briefly discussing the possible options for the treatment of Kaposi sarcoma after HSCT and when chemotherapy should be used.

Treatment option has been added and discussed under discussion heading.

-       I would recommend mentioning what other organs have been described to be involved in Kaposi sarcoma in addition to those found in this patient. I would also suggest discussing the case by Avivi et al. (2021, Leuk Lymphoma) in which a liver involvement was described and underlying the differences and similarities with the reported case.

Unfortunately, we did not find the case report of suggested Author Avivi et al. however, we have discussed our case more extensively with other case reports now.

Reviewer 2 Report

The Case Report Titled "Kaposi Sarcoma in a Child After Fanconi Anemia-Induced Haploidentical Hematopoietic Stem Cell Transplant: A Case Report" Authored by Mohammed Saud Alsaidan, Ohoud Zaid Aljarbou and Waleed Alajroush provides a rare case study of Kaposi Sarcoma following hematopopoetic stem cell transplantation. A case as that is important on the view of the possible molecular mechanisms that may lead to sarcoma development.
The study is well presented. The figures are informative and to the point. The report is well written and easy to follow. I would prefer that the conclusions would not start in the same manner as the discussion and the authors should consider editng the sentence in line 133.

Author Response

Dear Reviewer, 

Thank you for your valuable feedback. We have done all the suggested changes in the manuscript highlighted in yellow. 

Reviewer 2

The Case Report Titled "Kaposi Sarcoma in a Child After Fanconi Anemia-Induced Haploidentical Hematopoietic Stem Cell Transplant: A Case Report" Authored by Mohammed Saud Alsaidan, Ohoud Zaid Aljarbou and Waleed Alajroush provides a rare case study of Kaposi Sarcoma following hematopopoetic stem cell transplantation. A case as that is important on the view of the possible molecular mechanisms that may lead to sarcoma development.

The study is well presented. The figures are informative and to the point. The report is well written and easy to follow. I would prefer that the conclusions would not start in the same manner as the discussion and the authors should consider editng the sentence in line 133.

Thank you for the suggestion, conclusion has been edited.
